# Detecting Bacterial Cell Viability in Few µL Solutions from Impedance Measurements on Silicon-Based Biochips

**DOI:** 10.3390/ijms22073541

**Published:** 2021-03-29

**Authors:** Vinayak J. Bhat, Sahitya V. Vegesna, Mahdi Kiani, Xianyue Zhao, Daniel Blaschke, Nan Du, Manja Vogel, Sindy Kluge, Johannes Raff, Uwe Hübner, Ilona Skorupa, Lars Rebohle, Heidemarie Schmidt

**Affiliations:** 1Leibniz Institute of Photonic Technology, Albert-Einstein-Str. 9, 07745 Jena, Germany; vbhat204@gmail.com (V.J.B.); daniel.blaschke@leibniz-ipht.de (D.B.); uwe.huebner@leibniz-ipht.de (U.H.); 2Center for Microtechnologies, Chemnitz University of Technology, 09126 Chemnitz, Germany; mahdi.kiani@s2013.tu-chemnitz.de (M.K.); xianyue.zhao@enas.fraunhofer.de (X.Z.); 3Institute for Solid State Physics, Friedrich Schiller University Jena, Helmholtzweg 3, 07743 Jena, Germany; 4Helmholtz-Zentrum Dresden-Rossendorf, Bautzner Landstraße 400, 01328 Dresden, Germany; m.vogel@hzdr.de (M.V.); s.kluge@hzdr.de (S.K.); j.raff@hzdr.de (J.R.); i.skorupa@hzdr.de (I.S.); l.rebohle@hzdr.de (L.R.)

**Keywords:** cell viability, *Lysinibacillus sphaericus*, impedance biochips, membrane potential, dead bacterial cells, live bacterial cells

## Abstract

Using two different types of impedance biochips (PS5 and BS5) with ring top electrodes, a distinct change of measured impedance has been detected after adding 1–5 µL (with dead or live Gram-positive *Lysinibacillus sphaericus* JG-A12 cells to 20 µL DI water inside the ring top electrode. We relate observed change of measured impedance to change of membrane potential of *L. sphaericus* JG-A12 cells. In contrast to impedance measurements, optical density (OD) measurements cannot be used to distinguish between dead and live cells. Dead *L. sphaericus* JG-A12 cells have been obtained by adding 0.02 mg/mL of the antibiotics tetracycline and 0.1 mg/mL chloramphenicol to a batch with OD0.5 and by incubation for 24 h, 30 °C, 120 rpm in the dark. For impedance measurements, we have used batches with a cell density of 25.5 × 10^8^ cells/mL (OD8.5) and 270.0 × 10^8^ cells/mL (OD90.0). The impedance biochip PS5 can be used to detect the more resistive and less capacitive live *L. sphaericus* JG-A12 cells. Also, the impedance biochip BS5 can be used to detect the less resistive and more capacitive dead *L. sphaericus* JG-A12 cells. An outlook on the application of the impedance biochips for high-throughput drug screening, e.g., against multi-drug-resistant Gram-positive bacteria, is given.

## 1. Introduction

The knowledge of the growth state of bacteria in medically relevant samples is of importance since the bacterial growth is integral to infection [1]. Typically, to monitor cell viability after different drug treatment, an assay with viability dyes is chosen in dependence on the number and type of bacterial cells. Calibrated assays with single/double/triple staining can yield the percentage of live/dead/total bacterial cells. Intermediate bacterial cell states are assessed using double staining in combination with flow cytometry. However, the calibration of such assays depends on various chemical and biochemical aspects, e.g., on the metabolic behavior of bacterial cells, which is strongly dependent on the culture medium. High-throughput drug screening, e.g., to develop antibiotics against multi-resistant pathogens, relies on the exact detection of bacterial cell viability. Therefore, there is still a need to develop novel calibration methods for the determination of the viability of bacterial cells in different culture media, which also includes the determination of intermedial bacterial cell states.

Existing bacterial detection techniques are primarily divided into traditional culture methods, e.g., colony counting of diluted or whole sample solutions to determine bacterial concentration, which takes a long time to obtain results (24–48 h) and which needs an incubator, and biotechnological approaches, e.g., polymerase chain reaction amplification (PCR), fluorescence analysis, and traditional enzyme-linked immunoassay (ELISA) [2]. Gregori et al. [3] monitored the viability of *E. coli* cells from freshwater and seawater using the live-dead protocol developed by Barbesti et al. [4] based on simultaneous staining of bacteria nucleic acids by a permeant (SYBR Green I) and an impermeant (PI) fluorescent probe and the interpretation of the green versus the red fluorescence cytograms.

For the development of antibiotics, e.g., against *Staphylococcus aureus* and *Pseudomonas aeruginosa* [5], one usually exploits the fact that the bacterial cell membrane can be changed as a result of the treatment with antibiotics. Nowadays, bacterial cell viability is determined by fluorescence microscopy from live-dead (viability) staining. However, as pointed out by Stiefel et al. the accuracy of viability staining, e.g., of *S. aureus* cells, with SYTO9 and propidium iodide (PI) strongly depends on the accuracy in the PI signal [6]. There exist live/dead fixable dead cell stain kits on the market, which are based on the reaction of a fluorescent reactive dye with cellular proteins (amines). These dyes cannot penetrate live cell membranes, so only cell surface proteins are available to react with the dye, resulting in dim staining. The reactive dye can permeate the damaged membranes of dead cells and stain both the interior and exterior amines, resulting in more intense staining. This assignment may be accurate for cells with damaged cell membrane. However, cells with intact cell membrane do not necessarily be active because of the reduced cell membrane potential. In 2007 intermediate viability states have been observed on cells after medication only by combining results from flow cytometry and from live/dead fixable dead cell stain kits [7]. Robertson et al. optimized the protocol of LIVE/DEADR BacLightTM Bacterial Viability Kit for detecting viability of *E. coli* cells [8]. Bellali et al. studied the effect of atmospheric oxygen on the viability of gut microbes using flow cytometry and plating technique [9]. Freitas et al. investigated the viability of multi-resistant *Staphylococcus aureus* after antimicrobial photodynamic therapy with curcumin using fluorimetry [10]. Sun developed a novel colorimetric assay based on a Glucose oxidase/Horseradish peroxidase bienzyme system to detect viability of foodborne and drinking water bacterial infections, which is easily observable with the naked eye or recordable with a smartphone [11]. Franke et al. identified erythrosine as a cost-effective red dye with fluorescent properties for many Gram-positive and -negative bacteria [12]. Rosenberg et al. investigated the viability of adherent gram-positive *Staphylococcus epidermidis* and gram-negative *Escherichia* cells and pointed out that viability staining results of adherent cells should always be validated by an alternative method, preferably by cultivation [13]. Liao et al. optimized detectability of *E. coli* cells using fluorescence by using 3-(4,5-dimethylthiazol-2yl)-2,5-diphenyltetrazolium bromide (MTT) and phenazine methosulfate (PMS) as new reagents [2]. Ou et al. assessed four methods for viability analysis for *E. coli*, namely SYTO 9: propidium iodide fluorescence intensity ratio, an adjusted fluorescence intensity ratio, single-spectrum support vector regression (SVR) and multi-spectra SVR, and found that multi-spectra SVR is best suited [14].

Nowadays, high-throughput screening is usually carried out with fully automated laboratory automation systems with 384, 1536 or 3456 plates in order to save time and costs. The working volume of 384, 1536 or 3456 plates amounts to 25.0 µL, 1.5 µL, and 1.0 µL, respectively. As reviewed by Liao et al. [2] the biotechnological methods have many disadvantages including cost of materials (enzymes, antibodies, and antigens), low-temperature storage requirements, skilled and experienced operational needs, combination of different experimental methods with different instruments, and long and complicated operation methodology. Combining results from flow cytometry and from live/dead fixable dead cell stain kits is a too long and complicated operation methodology for online screening of bacterial cell viability, e.g., to determine intermediate states of bacterial cell viability [15] and to detect the long-term effects of drugs. In this work we present an approach, namely impedance spectroscopy with impedance biochips with ring top electrodes, that is suitable for fast screening of bacterial cell viability (1 min). As an example, we analyze the measured change of the complex, frequency-dependent impedance Z(ω) of the impedance biochip without and with up to 5 µL cell suspension with dead and live *Lysinibacillus sphaericus* JG-A12 cells added into the region of the ring top electrode with 20 µL DI water. We expect a different response of live and dead cells to the osmotic challenge in 20 µL DI water. Therefore, the corresponding change of membrane potential will not fully reflect the systematic change in the concentration of live and dead cells in the liquid added to 20 µL DI water. However, using experimental procedure the effect of the treatment of *L. sphaericus* JG-A12 with antibiotics on the membrane potential is much stronger in comparison to the effect of osmotic shock. We clearly reveal a more resistive and less capacitive impedance from live cells and a less resistive and more capacitive impedance from dead cells. Furthermore, we show that the real part of the effective impedance (ReZ(ω)) and that the imaginary part of the effective impedance can be related with the thermal movement and with the membrane potential of the cells, respectively. The paper is organized as follows: in Section 2 we present results from optical microscopy, optical density (OD) measurements, atomic force microscopy, plating measurements, and impedance measurements with impedance modeling (see more details in the Appendix A), in Section 3 we discuss the results and present a biological model relating the observed differences in the effective impedance of live and dead cells with their membrane potential and biological mobility. Section 4 presents the applied methods and materials. Section 5 closes with a summary.

## 2. Results

We investigated single cells and a set of up to 10^9^ cells/mL from a batch of *L. sphaericus* JG-A12 using standard methods (atomic force microscopy (Section 2.1), optical microscopy and optical density measurements, and plating experiments (Section 2.2) and by using the newly established impedance spectroscopy measurements on impedance biochips [16,17,18] (Section 2.3). Two different equivalent circuit models have been developed to extract the effective impedance change Z(ω) of the impedance chips after addition of 1, 2, 3, 4 and 5 µL of liquid with live cells and 1, 2, 3, 4 and 5 µL of liquid with dead cells to 20 µL DI water.

### 2.1. Optical Microscopy and Atomic Force Microscopy Measurements on L. sphaericus JG-A12 Cells

For optical microscopy 3 µL of cell suspension were dropped on a microscopy slide, covered by a cover slit and investigated in phase contrast mode using an Olympus BX61 microscope (Olympus, Hamburg, Germany). Using simple optical microscopy (phase contrast) it is not possible to distinguish between live and dead cells. Here, we show results from vision tests on *L. sphaericus* JG-A12 cells with a nominal OD2.0 (Figure 1a) and OD0.5 (Figure 1b) by optical microscopy imaging. A homogenous distribution of the rod-shaped *L. sphaericus* JG-A12 cells can be recognized. The cell density scales with the OD value. For the atomic force microscopy (AFM) experiments 10 microliter (µL) cell sample were added on freshly cleaved mica and incubated for 15 min to let cells attach to the mica surface. Remaining liquid was then removed by a filter paper and sample was dried in air for 30 min. Topography of a single *L. sphaericus* JG-A12 cell in air has been recorded by AFM measurements using an Asylum Research Cypher atomic force microscope (Asylum Research, Santa Barbara, CA, USA) in contact mode using OMCL TR400PSA cantilever (Olympus) with a resonance frequency of about 11 kHz and a stiffness of 0.02 Nm^−1^ (Figure 2). Shown *L. sphaericus* JG-A12 cell is collapsed in air during drying for 30 min because the inner cell pressure of 20 bar cannot be kept in air. The single cell in air is rod-shaped (height: ca. 300 nm, width: 1 µm, length: 3.5 µm) with a flattening at the base. For treated and dead *L. sphaericus* JG-A12 cells in DI water only a small loss of ca. 10% cell height is expected because in treated and dead *L. sphaericus* JG-A12 cells the tugor pressure [19] might get lower due to missing metabolic activity. The flagellae longer than 2–4 µm are clearly visible for the single *L. sphaericus* JG-A12 cell in air (Figure 2). In addition to the potential of the rod-shaped part of the *L. sphaericus* JG-A12 cell its flagellae may also influence the attachment to a surface.

### 2.2. Optical Density and Cell-Forming Units on L. sphaericus JG-A12 Cells

Plating experiments have been performed on the Live cell sample (0 h-treatment), Treated cell sample (3 h treatment), and on the Dead cell sample (24 h treatment) (Table 1), which have been resuspended in 50 mL DI water to adjust a final OD0.5 before (live cell sample) and after (Treated cell sample, Dead cell sample) medical treatment, respectively. The number of colony-forming units (CFU) has been determined by plating 100 µL of different cell dilutions from the corresponding samples on agar plates and by counting the CFU after incubation for 24 h. CFU amounts to 3.8 × 10^5^ CFU/mL, 6.1 × 10^4^ CFU/mL, and 2.94 × 10^3^ CFU/mL, respectively (Table 1). The cell plating experiments confirm that medical treatment further reduces the number of CFU. However, the suspension of the *L. sphaericus* JG-A12 cells in DI water reduces the number of CFU most significantly. This is expected, as water flooding with DI water inactivates the *L. sphaericus* JG-A12 cells. Optical density (OD) measurements have been performed on the Live cell sample, the Treated cell sample, and on the Dead cell sample (Table 1). For OD measurements, we used the UV-Vis-spectrometer Specord 50 (Analytik Jena, Jena, Germany). The results from OD measurements at 600 nm reveal that one cannot distinguish between cells from the Live cell sample, Treated cell sample, and Dead cell sample whose nominal OD amounts to 1.0 with a relative error of 1.0% 2.97 × 10^8^ cells/mL), 6.0% (2.83 × 10^8^ cells/mL), and 0.0% (3.00 × 10^8^ cells/mL), respectively (Table 1). For the impedance measurements, we have used a batch with OD8.5 and a batch with OD90.0 from the live cells and from the dead cells, which have been derived from a cell density of 25.5 × 10^8^ cells/mL and 270.0 × 10^8^ cells/mL, respectively. This indicates a nearly unchanged light scattering cross-section and light absorption [20] of live, treated, and dead cells for light with a wavelength of 600 nm. Unchanged light scattering cross-section hints towards an unchanged cell shape despite DI water flooding and medical treatment.

### 2.3. Impedance Spectroscopy on Impedance Biochips

Two types of impedance biochips BS5 and PS5 have been prepared for investigating the impedance variation of the biochip when adding live and dead *L. sphaericus* JG-A12 cells in DI water inside the ring electrode area. BS5 and PS5 are 1 × 1 cm^2^ large pieces from an implanted Si wafer with an Au ring top electrode with an area A = 0.22 cm^2^), a p^+^n junction (BS5) and a n^+^p junction (PS5), and with an unstructured Au bottom contact. The implantation parameters are given in Refs. [16,17,18]. Impedance has been recorded in the frequency range between 40 Hz and 1 MHz before filling (No fill) and after filling 20 µL DI water, and after subsequently adding 1, 2, 3, 4, and 5 µL liquid into the ring electrode area of the impedance biochips. Pipetting an additional 1 µL of liquid and performing the impedance spectroscopy typically lasts 1 min. A photograph of the impedance biochip before and after filling 20 µL DI water inside the ring electrode area is shown in Figure 3. To avoid sample evaporation and to realize a meaningful variation of the concentration of *L. sphaericus* JG-A12 cells, we have used a batch with OD8.5 (25.5 × 10^8^ cells/mL) and a batch with OD90.0 (270.0 × 10^8^ cells/mL) from the live cells and from the dead cells. The corresponding cell concentration is listed in Table 2. The cell concentration in 21 µL and in 25 µL liquid amounts to 1.2 × 10^8^ cells/mL to 24.7 × 10^8^ cells/mL, respectively. For a surface density larger 1.00 µm^−2^ it may be expected that the *L. sphaericus* JG-A12 cell touch each other. Measured impedance data are shown in Figure 3 and Figure 4. The absolute value of the complex, frequency-dependent impedance |Z| of the impedance biochip PS5 and BS5 ranges up to 10^3^ Ohm and 10^4^ Ohm, respectively. This difference in the absolute impedance value explains why the change of the impedance value after adding 20 µL and subsequently 1 µL, 1 µL, 1 µL, 1 µL, and 1 µL DI water, is better detectable for PS5 (Figure 4a) in comparison to BS5 (Figure 5a). Impedance biochip PS5 (Figure 4c) and BS5 (Figure 5b) reveal the strongest impedance changes when adding live cells and dead cells, respectively. It is clear that the absolute impedance depends on the impedance of the Au bottom electrode/Si, of the Si n^+^p junction (PS5) and Si p^+^n junction (BS5), of the Au ring top electrode/Si, and of the impedance of the liquid/Au and liquid/Si. Therefore, a clear relationship between the absolute impedance data and the concentration of live cells (see legend in Figure 4c) and of dead cells (see legend in Figure 5b) cannot be derived from change of impedance data in dependence on the total volume of liquid-filled inside the ring electron area.

### 2.4. Impedance Spectroscopy Modeling

In order to find the relationship between impedance of the impedance biochips and concentration of live and dead cells visible, we have calculated the effective impedance of the liquid/Au and liquid/Si by subtracting the modeled impedance of the Au bottom electrode/Si, of the Si n^+^p junction (PS5) and Si p^+^n junction (BS5), of the Au ring top electrode/Si from the measured total impedance (Figure 4 and Figure 5). The calculated effective impedance (Figure 6 and Figure 7) shows a clearer relationship between effective impedance and concentration of live cells (see legend Figure 8) and of dead cells (see legend in Figure 9). Because of the different order of total impedance, we used two different equivalent circuit models to describe the total impedance of PS5 and BS5 after adding a liquid. In this work, we link the capacitance of the ring top electrode C_Au/Si_, of the n^+^p junction C_n+p_ (PS5, Figure 6a) and of the p^+^n junction C_p+n_ (BS5, Figure 7a), and of the bottom contact C_Si/Au_ in the equivalent circuit diagram before adding a liquid (No fill) with physical parameters. The capacitance [21] of the ring top electrode C_Au/Si(n+)_ and C_Au/Si(p+)_ of PS5 and BS5, respectively, is calculated as follows:(1)$$C_{\mathrm{Au}/\mathrm{Si}(n+)}\;=\;A\cdot \sqrt{\left(q\varepsilon \varepsilon_{0}N_{D}\right)/2\Phi_{bi,n+}}$$
(2)$$C_{\mathrm{Au}/\mathrm{Si}(p+)}\;=\;A\cdot \sqrt{\left(q\varepsilon \varepsilon_{0}N_{A}\right)/2\Phi_{bi,p+}}$$

The junction capacitance of C_n+p_ and C_p+n_ of PS5 and BS5, respectively, is calculated as follows:(3)$$C_{n+p}\;=\;\frac{\varepsilon \varepsilon_{0}A}{x_{d,n^{+}p}}$$
(4)$$C_{p+n}\;=\frac{\varepsilon \varepsilon_{0}A}{x_{d,p^{+}n}}$$

The capacitance of the bottom electrode C_Si(p)/Au_ and C_Si(n)/Au_ of PS5 and BS5, respectively, is calculated as follows:(5)$$C_{\mathrm{Si}(p)/\mathrm{Au}}\;=\;A\cdot \sqrt{\left(q\varepsilon \varepsilon_{0}N_{A}\right)/2\Phi_{bi,p}}$$
(6)$$C_{\mathrm{Si}(n)/\mathrm{Au}}\;=\;A\cdot \sqrt{\left(q\varepsilon \varepsilon_{0}N_{D}\right)/2\Phi_{bi,n}}$$

The static dielectric constant of Si is ε=11.7. Values of the built-in potentials of the Au ring top electrode on PS5 $\Phi_{bi,n+}$ and on BS5${\;\Phi }_{bi,n}$, of the junction in PS5 $\Phi_{bi,n+p}$ and in BS5 $\Phi_{bi,p+n}$, and of the Au bottom electrode on PS5 $\Phi_{bi,p}$ and on BS5 $\Phi_{bi,n}$ are given in Table 2. The vacuum permittivity is $\varepsilon_{0}=8.854\cdot 10^{-14}F/\mathrm{cm}$. The effective area of unstructured bottom contact is limited by the area of structured counter electrode, i.e., by the area of the ring top electrode A. Therefore, we used the same value to describe the area A in Equations (1)–(6). The area of the ring electrode, which amounts to *A* = 0.22 cm^2^. We have modeled the impedance of PS5 and BS5 before filling (No fill) and with 20 µL DI water filling using the equivalent circuit model shown in Figure 6a and Figure 7a, respectively. The calculated values of the capacitance of ring electrode, junction capacitance, and capacitance of bottom contact (Table 1) have been taken as a starting point for the modeling before filling (No fill). Measured (red symbols) and modeled (red solid lines) impedance data of the three PS5 chips and of the three BS5 chips are shown in Figure A1 and in Figure A2, respectively. The extracted values of the equivalent circuit model for PS5 and BS5 before filling (No fill) are listed in Table A1 and in Table A2, respectively. The impedance data of PS5 and BS5 after filling with 20 µL DI water have also been modeled with the equivalent circuit model shown in Figure 6a and Figure 7a, respectively. Using the experimental impedance data after filling with 20 µL DI water, we have readjusted the impedance parameters of the impedance bottom contact, of the junction, and of the ring electrode modeled before filling (No Fill) to obtain the impedance parameters after filling (20 µL DI). The impedance of the bottom contact increases because of the effective increase of top electrode size after filling liquid inside the region of the ring top electrode. The comparison of the modeled equivalent circuit parameters (No Fill and 20 µL DI) reveals that for the three PS5 chips (DI water, Dead, Live in Table A1) and for the three BS5 chips (DI water, Dead, Live in Table A2) mainly C_Au/liq_, R_Au/liq_, R_liq_, and C_j_ changed. In addition, for the three PS5 chips R_pn_ and R_Si(n)/Au_ changed (Table A1) and for the BS5 chips (DI water, Dead) R_j_ changed (Table A2) and for the BS5 chip (Live) C_j_ and R_p+n_ changed (Table A2). Finally, the impedance data of PS5 and BS5 after adding 1, 2, 3, 4, and 5 µL liquid to the 20 µL DI water have been modeled by adding a complex frequency-dependent impedance Z(ω) in parallel (Figure 6b) and in series (Figure 7b) to the impedance of the top electrode. This effective impedance Z(ω) of Live cells and of Dead cells has been constructed by successively subtracting all the components of the equivalent circuit (except Z(ω)) shown in Figure 6b and Figure 7b from the measured total impedance on PS5 and BS5, respectively. The extracted effective impedance of PS5 and of BS5 is plotted as Re Z(ω) and as -Im Z(ω) in Figure 8 and Figure 9, respectively. As expected, for PS5 and for BS5 the effective impedance does not change significantly after adding 1, 2, 3, 4, and 5 µL DI water (blue lines in Figure 8 and Figure 9). We expect a different response of live and dead cells to the osmotic challenge in 20 µL DI water. The corresponding change of membrane potential will not fully reflect the systematic change in the concentration of live and dead cells in the liquid added to 20 µL DI water. It is clearly visible that the effective impedance of PS5 changes not fully/fully systematically when 1, 2, 3, 4, and 5 µL liquid with dead/live cells are added to the 20 µL DI water (Figure 8). Also, it is clearly visible that the effective impedance of BS5 changes nearly/not fully systematically when 1, 2, 3, 4, and 5 µL liquid with dead/live cells are added to the 20 µL DI water (Figure 9). Therefore, we can only conclude that the effective impedance of PS5 and of BS5 changes nearly systematically when 1, 2, 3, 4, and 5 µL liquid with live cells (green lines in Figure 8) and with dead cells (black lines in Figure 9) is added to the 20 µL DI water.

## 3. Discussion

The systematic change in the effective impedance of PS5 (Z(ω) in parallel with surface impedance formed by the top contact, Figure 6b, Table A1) when 1, 2, 3, 4, and 5 μL liquid with live cells are added hint toward a change of the effective impedance, which is more resistive (Re Z(ω)) and less capacitive (-Im Z(ω)). Also, the nearly systematic change in the effective impedance of BS5 (Z(ω) in series with surface impedance formed by the top contact, Figure 7b, Table A2) when 1, 2, 3, 4, and 5 μL liquid with dead cells are added hint toward a change of the effective impedance, which is less resistive (Re Z(ω)) and more capacitive (-Im Z(ω)). In the following, we present a model that relates the observed impedance change with possible differences between the impedance of live and dead *L. sphaericus* JG-A12 cells. The model relies on the assumption that live *L. sphaericus* JG-A12 cells (Figure 10a) are more resistive and less capacitive, and that dead *L. sphaericus* JG-A12 cells (Figure 10c) are less resistive and more capacitive. Live cells are more mobile than dead cells. Figure 10 shows a sketch of a cross-section through the impedance biochip with live, treated, and dead *L. sphaericus* JG-A12 cell in DI water attached to the surface of the impedance biochip PS5 or BS5. The cell wall of the gram-positive *L. sphaericus* JG-A12 cells has to deal with inner cell pressure of 20 bar (2–5 bar for gram-negative). So, even if the tugor pressure [19] in dead *L. sphaericus* JG-A12 cells is getting lower due to missing cell metabolic activity, the cell skeleton should stay in shape more or less. We estimated a loss of height of approx. 10% when comparing live (Figure 10a) and dead (Figure 10c) *L. sphaericus* JG-A12 cells in DI water. Due to heat movements, live cells are expected to constantly hit each other from all directions and thus to show a larger ac resistance. Furthermore, live cells are expected to have an unchanged membrane potential (red thick line surrounding the live cell, Figure 10a) which screens the charges inside the live cell and reduces the polarizability of live cells. This explains why live cells are more resistive and less capacitive. Dead cells cannot follow heat movements. They are expected to stick more to the surface of the impedance biochip. The membrane potential (no red line surrounding dead cells, Figure 10c) is strongly reduced, and charges inside the dead cell can follow the ac electric field applied during impedance measurements (f_ac_ = 40 Hz to 1 MHz, V_ac_ = 50 mV). This explains why dead cells are less resistive and more capacitive. Presented effective impedance data prove our working hypotheses. We have shown that one can measure the viability of *L. sphaericus* JG-A12 cells using two types of impedance biochips. The impedance biochip PS5 with the smaller total impedance (up to 0.4 kΩ) can be used to detect an effective, more resistive, and less capacitive impedance from live cells in a liquid in parallel to the surface impedance (Figure 10a). Also, the impedance biochip BS5 with the larger total impedance (up to 7 kΩ) can be used to detect an effective, less resistive and more capacitive impedance from dead cells in a liquid in series to the surface impedance (Figure 10c). In the next step, we will investigate how intermediate levels of viability can be detected, e.g., by analyzing impedance data from so-called treated cells (Figure 10b) which have been recorded on PS5 and BS5. In this work, we used one type of antibiotics with a fixed concentration (0.02 mg/mL tetracycline, 0.10 mg/mL chloramphenicol) and two incubation times (3 h, 24 h) to block the peptide biosynthesis in *L. sphaericus* JG-A12 cells. In order to find new drugs against multi-resistant pathogens, high-throughput screening of antibiotics (type, concentration, incubation time) with the impedance biochips may be envisioned.

## 4. Materials and Methods

In this work, we used *L. sphaericus* JG-A12 cells which have been isolated from the uranium mining waste pile “Haberland” near Johanngeorgenstadt in Saxony, Germany. The cells were cultivated in nutrient broth (8 g/L, Mast Group) overnight in Erlenmeyer flasks at 30 °C and shaking at 120 rpm. Correlation between optical density at 600 nm (OD_600_) and cell number was achieved by cell counting under a microscope using a Neubauer counting chamber.

We performed atomic force microscopy (AFM), optical microscopy, OD measurements, plating experiments, and impedance spectroscopy measurements to analyze the viability of *L. sphaericus* JG-A12 in DI water before and after treatment with antibiotics. Here, in this work, we split the cell culture into three samples (Live, Treated (3 h-treated), Dead (24 h-treated)), and cells were harvested by centrifugation. After washing steps with DI water cells were resuspended in 50 mL DI water to adjust a final OD_600_ of 0.5. Washing and suspension in DI water strongly affects the number of colony-forming units because osmotic pressure can cause cell lysis. In order to keep the water flooding and possible cell bursting comparable, before performing the plating experiments and the impedance measurements, we suspended live and dead cells for 24 h in DI water before performing the plating experiments and the impedance measurements, respectively. To block the peptide biosynthesis and to obtain dead cells 0.02 mg/mL the antibiotics tetracycline and 0.1 mg/mL chloramphenicol were added to one sample and were incubated for 3 h (Treated cells) and for 24 h (Dead cells), 30 °C, 120 rpm in the dark. Before further use, treated cells (3 h) and dead cells (24 h) were washed two times and resuspended in DI water.

Two types of impedance biochips BS5 and PS5 have been prepared for investigating the impedance variation of the biochip when adding 1 µL (OD8.5), 1 µL (OD8.5), 1 µL (OD8.5), 1 µL (OD90.0), 1 µL (OD90.0) liquid with live and dead *L. sphaericus* JG-A12 cells in DI water to 20 µL DI water. The corresponding cell concentration is listed in Table 2. For fabricating the BS5 biochips, Boron ions (B+) implanted into 4 inch Si:P wafers and for creating the PS biochips, phosphorous (P−) ions implanted into 4 inch Si:B wafers (Table 2). After implantation, the 4 wafers have been cut into 1 × 1 cm^2^ large pieces, and one 150 nm thick gold (Au) ring top electrode with inner and outer diameters of 5.7 mm and 7.8 mm and an unstructured, 1 × 1 cm^2^ large Au bottom electrode have been deposited on every 1 × 1 cm^2^ large piece by dc-magnetron sputtering. The absolute value of the complex, frequency-dependent impedance |Z| of the impedance biochips ranges up to 10^5^ Ohm. The frequency-dependent real part (Re Z) and imaginary part (Im Z) of the complex impedance has been recorded before and after filling the inner ring region of the ring top electrode in the frequency range from 40 Hz to 1 MHz under illumination at room temperature by using the Agilent 4294A precision impedance analyzer. The Agilent 4294A can measure the impedance of the device under test in the impedance range from 10^−3^ to 10^+8^ Ohm with a frequency resolution of 1 MHz. We analyzed the real and imaginary part of the measured frequency-dependent impedance data of the impedance biochip before and after filling the inner ring region of the ring top electrode using an equivalent circuit model. The equivalent circuit model of the impedance biochip before filling is a physical model and uses calculated capacitance values to describe the capacitance of the unstructured bottom electrode C_Si/Au_, the junction capacitance C_n+p_ (PS5) and C_p+n_ (BS5), and the capacitance of the ring top electrode C_Au/Si_. The impedance data of PS5 and BS5 after adding 1, 2, 3, 4, and 5 µL liquid to the 20 µL DI water have been modeled by adding a complex frequency-dependent impedance Z(ω) in parallel and in series to the impedance of the top electrode. This effective impedance Z(ω) has been determined by subtracting the components of the equivalent circuit with 20 µL DI water from the measured total impedance. Finally, the systematic change of the effective impedance when 1, 2, 3, 4, and 5 µL liquid with live cells and dead cells are added to 20 µL DI water is analyzed in terms of resistance (Re Z(ω)) and capacitance (−Im Z(ω)).

## 5. Conclusions

In this work, we have presented impedance spectroscopy with impedance biochips with ring top electrodes for the fast screening of bacterial cell viability (1 min). We have analyzed the viability of *L. sphaericus* JG-A12 cells in 1, 2, 3, 4, and 5 µL DI water added to 20 µL DI water without *L. sphaericus* JG-A12 cells using two types of impedance biochips (PS5 and BS5). The impedance biochip PS5 with the smaller total impedance (up to 0.4 kΩ) can be used to detect an effective, more resistive, and less capacitive impedance from live cells in a liquid in parallel to the surface impedance. Also, the impedance biochip BS5 with the larger total impedance (up to 7 kΩ) can be used to detect an effective, less resistive, and more capacitive impedance from dead cells in a liquid in series to the surface impedance. We developed a model that relates observed differences in the effective impedance of live and dead *L. sphaericus* JG-A12 cells with their different heat movement and membrane potential. In the future, intermediate levels of bacterial cell viability will be analyzed by determining effective impedance of treated cells using both types of impedance biochips (PS5 and BS5) and methodology for online screening of bacterial cell viability using impedance spectroscopy with presented impedance biochips will be developed.

## Figures and Tables

**Figure 1 ijms-22-03541-f001:**
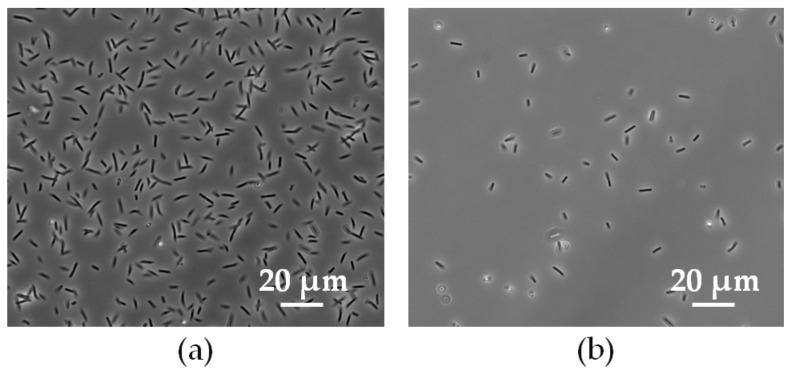
215 µm × 165 µm large optical microscopy image of the gram-positive *L. sphaericus* JG-A12 cells before treatment with antibiotics with (**a**) OD2.0 (6.0 × 10^8^ cells/mL) and with (**b**) OD0.5 (1.5 × 10^8^ cells/mL) before plating, scale bar 20 µm.

**Figure 2 ijms-22-03541-f002:**
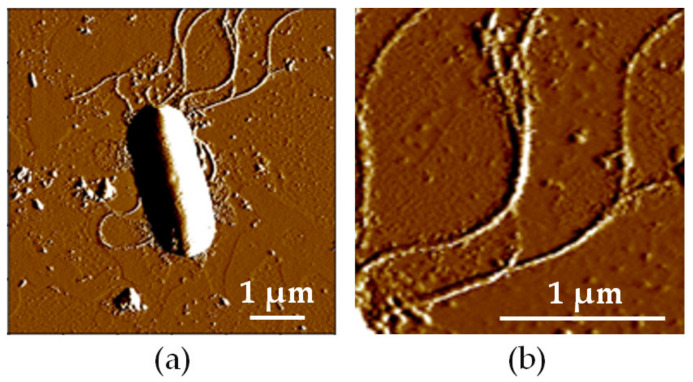
(**a**) 6 × 6 µm^2^ and (**b**) 2 × 2 µm^2^ large atomic force microscopy (AFM) image of a single gram-positive *L. sphaericus* JG-A12 cell in air, scale bar 1 µm. (**b**) The flagellae are clearly visible. However, shown *L. sphaericus* JG-A12 cell is collapsed in air because the inner cell pressure of 20 bar cannot be kept in air. For treated and dead *L. sphaericus* JG-A12 cells in DI water, only a small loss of ca. 10% height is expected because in treated and dead *L. sphaericus* JG-A12 cells the tugor pressure might get lower due to missing metabolic activity.

**Figure 3 ijms-22-03541-f003:**
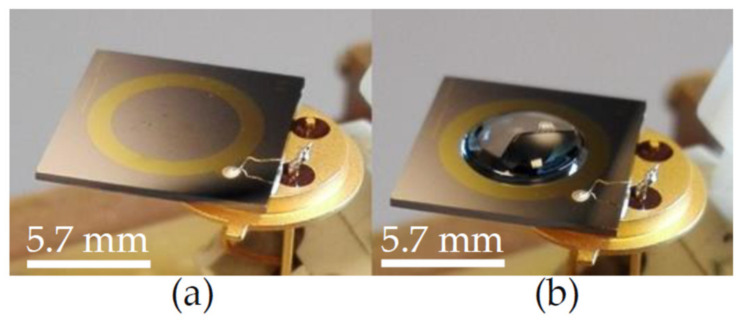
Photograph of an impedance biochip with Au ring electrode wire-bonded to a diode socket (**a**) before and (**b**) after filling 20 µL DI water inside the top electrode area. The outer and inner diameter of the ring electrode amounts to 5.7 mm and 7.8 mm, respectively. Courtesy of HZDR Innovation GmbH (https://hzdr-innovation.de/en/products/impedance-biochip/).

**Figure 4 ijms-22-03541-f004:**
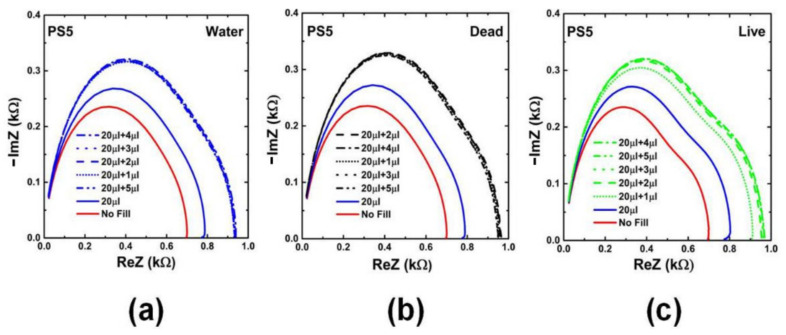
Nyquist plots for measured impedance of impedance chip PS5 (Table A1) without filling (No fill, red solid line), with filling with DI water (20 μL, blue solid line), and with addition of 1 μL (OD8.5) (20 μL + 1 μL), 2 μL (OD8.5) (20 μL + 2 μL), 3 μL (OD8.5) (20 μL + 3 μL), 4 μL (OD90.0) (20 μL + 4 μL), and 5 μL (OD90.0) (20 μL + 5 μL) (**a**) DI water (blue scattered lines) (**b**) Dead cells (black scattered lines) and (**c**) Live cells (green scattered lines). The corresponding concentration of live and dead cells is given in Table 1.

**Figure 5 ijms-22-03541-f005:**
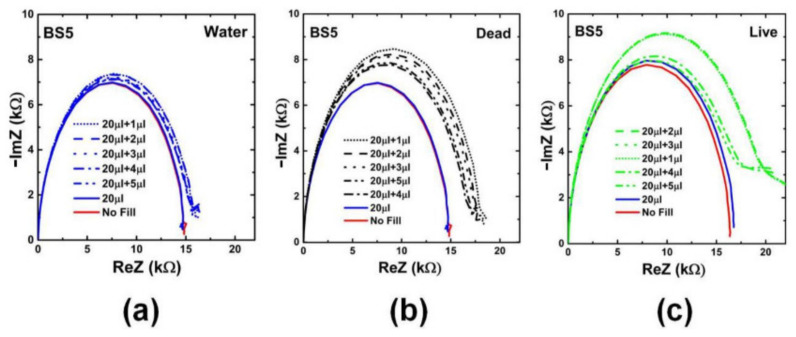
Nyquist plots for measured impedance of impedance chip BS5 (Table A2) without filling (No fill), with filling with DI water (20 μL), and with addition of 1 μL (OD8.5) (20 μL + 1 μL), 2 μL (OD8.5) (20 μL + 2 μL), 3 μL (OD8.5) (20 μL + 3 μL), 4 μL (OD90.0) (20 μL + 4 μL), and 5 μL (OD90.0) (20 μL + 5 μL) (**a**) DI water (**b**) Dead cells and (**c**) Live cells. The corresponding concentration of live and dead cells is given in Table 1.

**Figure 6 ijms-22-03541-f006:**
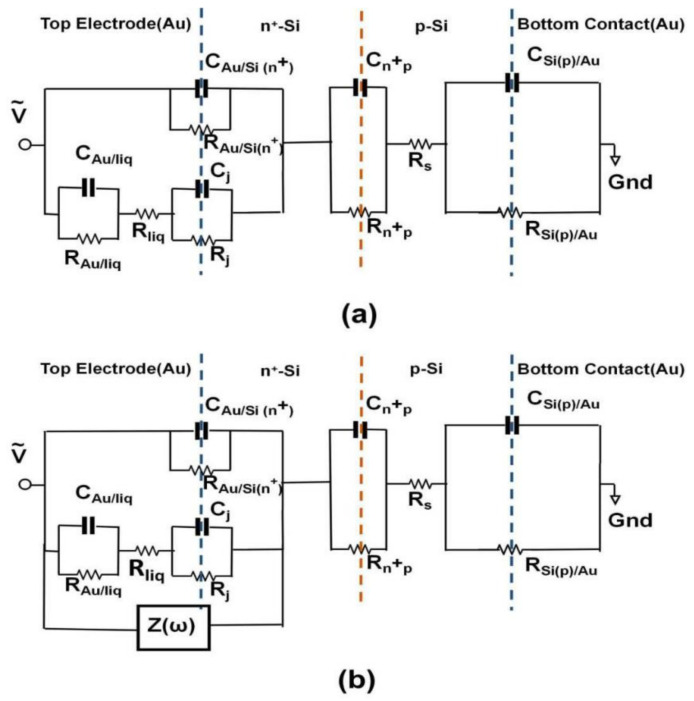
Equivalent circuit model of PS5 for (**a**) No Fill and 20 μL Dl. The changes in impedance (Figure 4) after addition of 1 μL, 2 μL, 3 μL, 4 μL, and 5 μL into the region of the ring top electrode are extracted by adding (**b**) Z(ω) in parallel to the ring electrode impedance components of 20 μL DI (**a**). Modeled values of RC pairs are listed in Table A1. The left blue, orange, and right blue dotted line indicates the interface between Au ring electrode and Si(n+), the position of n^+^p junction and the interface between Si(p) and Au bottom electrode, respectively.

**Figure 7 ijms-22-03541-f007:**
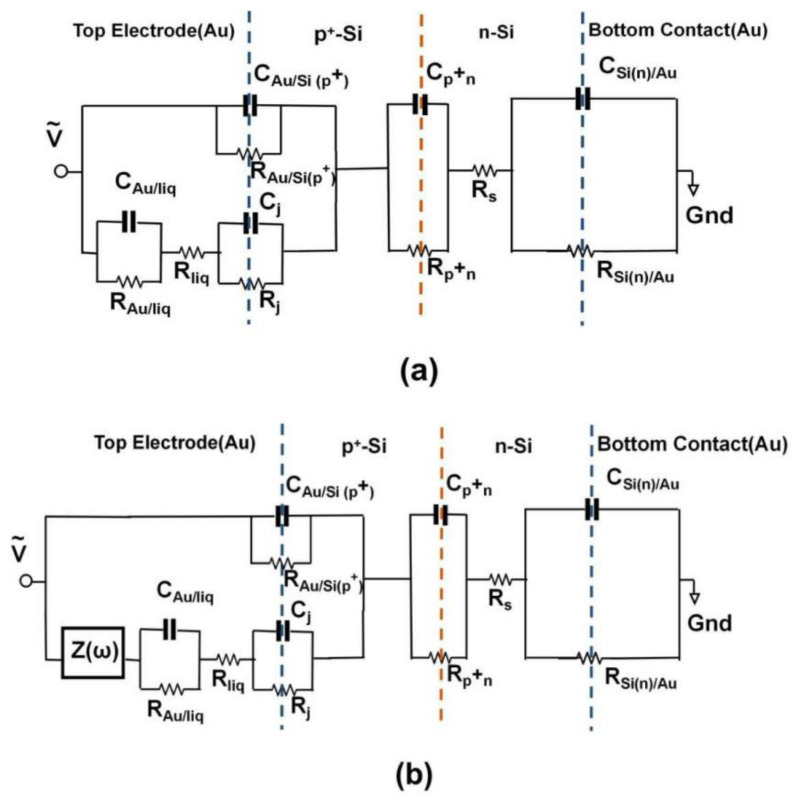
Equivalent circuit model of BS5 for (**a**) No Fill and 20 μL DI water. The changes in impedance (Figure 5) after addition of 1 μL, 2 μL, 3 μL, 4 μL, and 5 μL into the region of the ring top electrode are extracted by adding (**b**) Z(ω) in series to the ring electrode impedance components of 20 μL DI (**a**). Modeled values of RC pairs are listed in Table A2. The left blue, orange, and right blue dotted line indicates the interface between Au ring electrode and Si(p+), the position of p^+^n junction and the interface between Si(n) and Au bottom electrode, respectively.

**Figure 8 ijms-22-03541-f008:**
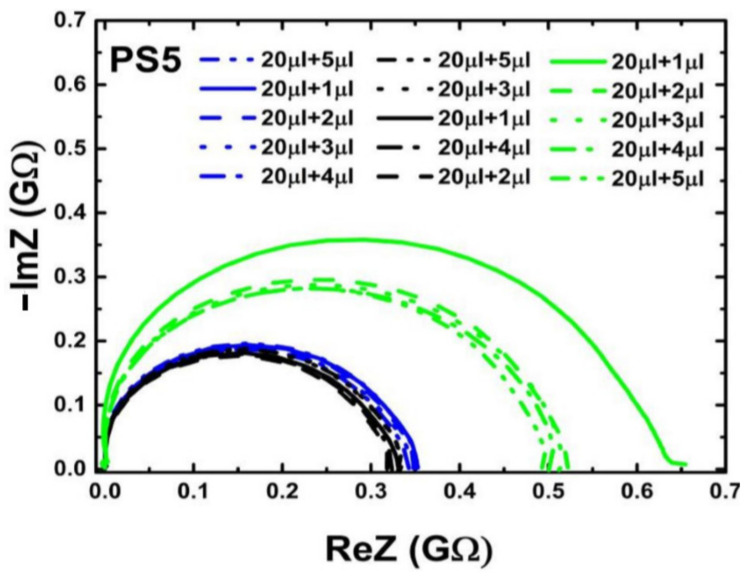
Nyquist plot of extracted effective impedance (Z(ω)) from the equivalent circuit model Figure 6b for PS5. The extracted effective impedance is represented for dead (black lines), for live (green lines), and for DI water sample (blue lines). Not fully systematic/systematic change of effective impedance (ReZ and Im Z) when adding liquid with dead/live cells to 20 µL DI water is visible.

**Figure 9 ijms-22-03541-f009:**
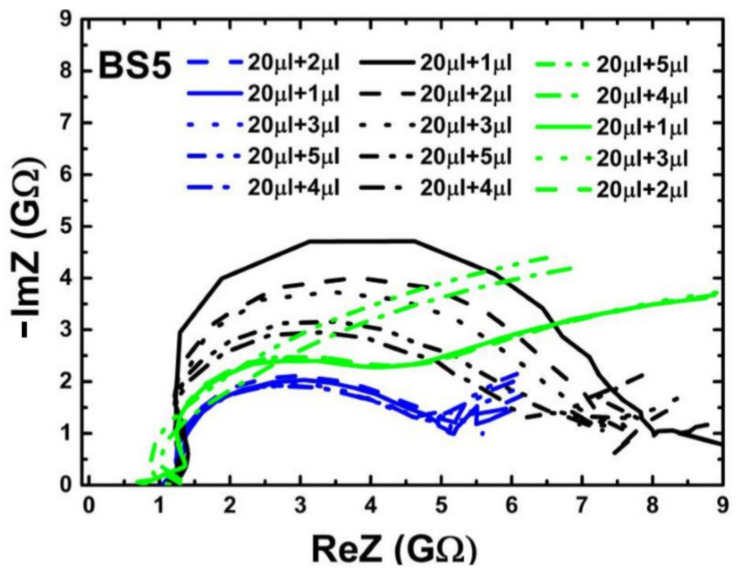
Nyquist plot of extracted effective impedance (Z(ω)) from the equivalent circuit model Figure 7b for BS5. The extracted effective impedance is represented for dead (black lines), for live (green lines), and for DI water sample (blue lines). Nearly systematic/not fully systematic change of effective impedance (ReZ and Im Z) when adding liquid with dead/live cells to 20 µL DI water is visible.

**Figure 10 ijms-22-03541-f010:**
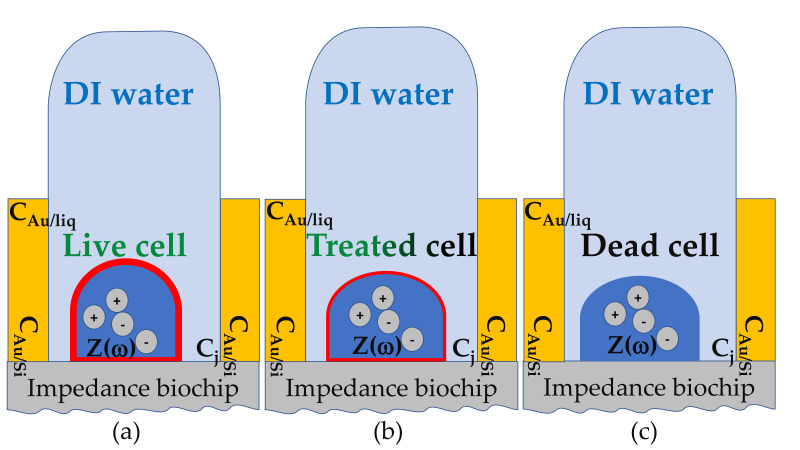
Schematic sketch of a cross-section through the impedance biochip with a (**a**) live, (**b**) treated, and (**c**) dead *L. sphaericus* JG-A12 cell in DI water attached to the surface of the impedance biochip. The effective impedance Z(ω) of the cell is added in parallel and in series to the PS5 and BS5 impedance biochip, respectively. The height of the cross-section of the Au ring top electrode (yellow) amounts to 150 nm. The height of 25 µL liquid (light blue) inside the region of the Au ring top electrode amounts to ca. 1 mm (Figure 3b). The water drop is held on the underside by the approximately 150 nm high Au ring top electrode and prevented from leaking. The place of origin of capacitance of the ring top electrode C_Au/Si_, of the capacitance between ring top electrode and liquid C_Au/liq_, of capacitance between liquid and surface of impedance biochip C_j_ and of the effective impedance of the *L. sphaericus* JG-A12 cell Z(ω) is also indicated.

**Table 1 ijms-22-03541-t001:** Concentration of *L. sphaericus JG-A12* cells after adding 0.02 mg/mL the antibiotics tetracycline and 0.1 mg/mL chloramphenicol and after incubation for 3 h and 24 h, 30 °C, 120 rpm in the dark.

Cell Concentration	0 h Treatment Live Cell Sample	3 h Treatment Treated Cell Sample	24 h Treatment Dead Cell Sample
Overall cells [10^8^ cells/mL]	2.97000	2.83000	3.00000
Living cells [10^8^ CFU/mL]	0.00380	0.00061	0.00003
Percentage of living cells [%]	0.130	0.020	0.001

No medical treatment has been applied to the Live cell sample “0 h treatment”. The concentration of overall cells in units of 10^8^ cells/mL and of living cells in units of 10^8^ CFU/mL has been determined using optical density measurements and plating experiments, respectively. For plating we used 100 µL of different cell dilutions from the Live, Treated, and Dead cell samples on agar plates and counting colony-forming units (CFU) after incubation for 24 h. For the impedance spectroscopy, we used the Live cell sample and the Dead cell sample from batches with a cell density of 25.5 × 10^8^ cells/mL (OD8.5) and 270.0 × 10^8^ cells/mL (OD90.0).

**Table 2 ijms-22-03541-t002:** Test volume inside ring electrode area in units of µL, concentration of *L. sphaericus* JG-A12 cells inside ring electrode area of the impedance chips with 20 µL DI water and with subsequently 1 µL (OD8.5), 1 µL (OD8.5), 1 µL (OD8.5), 1 µL (OD90.0) and 1 µL (OD90.0) liquid added and corresponding number of cells in test volume and in units of 10^4^ cells. OD1.0 corresponds to 3.0 × 10^8^ cells/mL.

Liquid Added	20 µL DI Water	1 µL (OD8.5)	1 µL (OD8.5)	1 µL (OD8.5)	1 µL (OD90.0)	1 µL (OD90.0)
Test volume inside ring electrode area (µL)	20	21	22	23	24	25
Cell concentration (10^8^ cells/Ml)	0.0	1.2	2.3	3.3	14.4	24.7
Number of cells in the test volume ^1^ (10^4^ cells)	0	252	506	759	3456	6175
Cell surface density ^2^ (µm^−2^)	0.00	0.10	0.20	0.29	1.35	2.42

^1^ Cell concentration has been multiplied with total volume of liquid added. ^2^ The surface area corresponds to the size of the inner region of the ring top electrode (25.51 mm^2^ = 2.551 × 10^7^ μm^2^).

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
