# Peer review of "Detecting Bacterial Cell Viability in Few µL Solutions from Impedance Measurements on Silicon-Based Biochips"

_ijms, 2021, doi:10.3390/ijms22073541_

Round 1

Reviewer 1 Report

The authors report the development of impedance-based biochips for the detection of bacterial cell viability.

As a general appreciation, the topic is highly sound and results of interest for the scientific community, even though I have doubts that the IJMS journal is the most suited to publish this work.

In my opinion the manuscript content could be improved in its organization, namely in the following aspects:

  • The introduction lacks a storyline. The authors make an extensive enumeration of studies on bacterial cell viability (line 77-97) but without pointing out their limitations, performance, how those do not respond to the current needs and how they compare with the present work. A general comment about those approaches, shows up later in the text, and only after covering the biotechnological methods a bit out of order. A more clear introduction to the advantages of the present work is missing at this point.
  • The description of the principle of functioning of the detection system should be introduced earlier at the Introduction/Results section rather than only at the Discussion section.
  • In the discussion the authors could try to present an explanation to the fact that the system changes nearly systematically (not fully systematic) when increasing volume of liquid with dead cells is added to the 20 μl DI water (Fig. 8).

Author Response

Reviewer 1 (Submission Date 27 February 2021, Date of this review 17 Mar 2021 20:04:55):

Comments and Suggestions for Authors

The authors report the development of impedance-based biochips for the detection of bacterial cell viability. As a general appreciation, the topic is highly sound and results of interest for the scientific community, even though I have doubts that the IJMS journal is the most suited to publish this work. In my opinion the manuscript content could be improved in its organization, namely in the following aspects:

  • The introduction lacks a storyline. The authors make an extensive enumeration of studies on bacterial cell viability (line 77-97) but without pointing out their limitations, performance, how those do not respond to the current needs and how they compare with the present work. A general comment about those approaches, shows up later in the text, and only after covering the biotechnological methods a bit out of order. A more clear introduction to the advantages of the present work is missing at this point.

Answer: We have elaborated the storyline in the beginning of the introduction and revised the manuscript as follows:

The knowledge of the growth state of bacteria in medically relevant samples is of importance since the bacterial growth is integral to infection [1]. High-throughput drug screening, e.g. to develop antibiotics against multiresistent pathogens, relies on the detection of bacterial cell viability.

-->

The knowledge of the growth state of bacteria in medically relevant samples is of importance since the bacterial growth is integral to infection [1]. Typically, to monitor cell viability after different drug treatment, an assay with viability dyes is chosen in dependence on the number and type of bacterial cells. Calibrated assays with single/double/triple staining can yield the percentage of live/dead/total bacterial cells. Intermediate bacterial cell states are assessed using double staining in combination with flow cytometry. However, the calibration of such assays depends on various chemical and biochemical aspects, e.g. on the metabolic behaviour of bacterial cells which is strongly depending on the culture medium.  High-throughput drug screening, e.g. to develop antibiotics against multiresistent pathogens, relies on the exact determination of bacterial cell viability. Therefore, there is still a need to develop novel calibration methods for the determination of the viability of bacterial cells in different culture media which also includes the determination of intermedial bacterial cell states.

  • The description of the principle of functioning of the detection system should be introduced earlier at the Introduction/Results section rather than only at the Discussion section.

Answer: We have introduced the principle of functioning of the detection system in the Introduction of revised manuscript as follows:

We investigated single cells and a set of up to 109 cells/ml from a batch of L. sphaericus JG-A12 using standard methods (atomic force microscopy (Sect. 2.1), optical microscopy and optical density measurements, and plating experiments (Sect. 2.2) and by using the newly established impedance spectroscopy measurements on impedance biochips [16-18] (Sect. 2.3).

-->

We investigated single cells and a set of up to 109 cells/ml from a batch of L. sphaericus JG-A12 using standard methods (atomic force microscopy (Sect. 2.1), optical microscopy and optical density measurements, and plating experiments (Sect. 2.2) and by using the newly established impedance spectroscopy measurements on impedance biochips [16-18] (Sect. 2.3). Two different equivalent circuit models have been developed to extract the effective impedance change Z(w) of the impedance chips after addition of 1, 2, 3, 4 and 5 µl of liquid with live cells and 1, 2, 3, 4 and 5 µl of liquid with dead cells to 20 µl DI water.

  • In the discussion the authors could try to present an explanation to the fact that the system changes nearly systematically (not fully systematic) when increasing volume of liquid with dead cells is added to the 20 μl DI water (Fig. 8).

Answer: We fully agree with the reviewer that it is very important to discuss nearly systematic and not fully systematic change of effective impedance when increasing volume of liquid with dead cells or live cells is added to 20 µl DI water. We revised caption of Fig. 7 and of Fig. 8 and added a discussion of systematic/not fully systematic change of effective impedance in the revised manuscript.

Fig. 7. Nyquist plot of extracted effective impedance (Z(ω)) from the equivalent circuit model Fig. 5b for PS5. The extracted effective impedance is represented for dead (black lines), for live (green lines), and for DI water sample (blue lines).

-->

Fig. 7. Nyquist plot of extracted effective impedance (Z(ω)) from the equivalent circuit model Fig. 5b for PS5. The extracted effective impedance is represented for dead (black lines), for live (green lines), and for DI water sample (blue lines). Not fully systematic/systematic change of effective impedance (ReZ and Im Z) when adding liquid with dead/live cells to 20 µl DI water is visible.

Fig. 8. Nyquist plot of extracted effective impedance (Z(ω)) from the equivalent circuit model Fig. 6b for BS5. The extracted effective impedance is represented for dead (black lines), for live (green lines), and for DI water sample (blue lines).

-->

Fig. 8. Nyquist plot of extracted effective impedance (Z(ω)) from the equivalent circuit model Fig. 6b for BS5. The extracted effective impedance is represented for dead (black lines), for live (green lines), and for DI water sample (blue lines). Nearly systematic/not fully systematic change of effective impedance (ReZ and Im Z) when adding liquid with dead/live cells to 20 µl DI water is visible.

It is clearly visible that the effective impedance of PS5 changes systematically when 1, 2, 3, 4, and 5 µl liquid with live cells are added to the 20 µl DI water (green lines in Fig. 7). Please note that the effective impedance changes in the order 20 µl + 1 µl, 20 µl + 2 µl, 20 µl + 3 µl, 20 µl + 5 µl, and 20 µl + 4 µl (legend with black lines in Fig. 8). Therefore, we can only conclude that the effective impedance of BS5 changes nearly systematically when 1, 2, 3, 4, and 5 µl liquid with dead cells is added to the 20 µl DI water (black lines in Fig. 8).

-->

We expect a different response of live and dead cells to the osmotic challenge in 20 µl DI water. The corresponding change of membrane potential will not fully reflect the systematic change in the concentration of live and dead cells in the liquid added to 20 µl DI water. It is clearly visible that the effective impedance of PS5 changes not fully/fully systematically when 1, 2, 3, 4, and 5 µl liquid with dead/live cells are added to the 20 µl DI water (Fig. 7). And it is clearly visible that the effective impedance of BS5 changes nearly/not fully systematically when 1, 2, 3, 4, and 5 µl liquid with dead/live cells are added to the 20 µl DI water (Fig. 8). Therefore, we can only conclude that the effective impedance of PS5 and of BS5 changes nearly systematically when 1, 2, 3, 4, and 5 µl liquid with live cells (green lines in Fig. 7) and with dead cells (black lines in Fig. 8) is added to the 20 µl DI water.

Reviewer 2 Report

The submitted article is very interesting, carefully prepared, and contains many valuable results. however, I would like to point out a few minor items that should be corrected or commented on. 
1) The use of deionized water in this study is controversial. Doesn't the osmotic shock, which certainly occurred, affect cell viability? 
2) Under what conditions were the cells centrifuged (too much centrifugal acceleration could also damage the cells)?
3) On what basis is the cell count (CFU) reported? What method was used?
4) There are Latin names of microorganisms in the text that are not in italics.
5) A photo of the measuring system (measuring cell with electrode) would be a valuable addition.

Author Response

Comments and Suggestions for Authors (Submission Date 27 February 2021, Date of this review 18 Mar 2021 10:33:28)

The submitted article is very interesting, carefully prepared, and contains many valuable results. However, I would like to point out a few minor items that should be corrected or commented on. 

1) The use of deionized water in this study is controversial. Doesn't the osmotic shock, which certainly occurred, affect cell viability? 

Answer: Yes, the osmotic shock will affect cell viability. We expect a different response of live and dead cells to the osmotic challenge in 20 µl DI water. Therefore, the corresponding change of membrane potential will not fully reflect the systematic change in the concentration of live and dead cells in the liquid added to 20 µl DI water. However, using described experimental procedure the effect of the treatment of L. sphaericus JG-A12 with antibiotics on the membrane potential is much stronger in comparison to the effect of osmotic shock. We clearly reveal a more resistive and less capacitive impedance from live cells and a less resistive and more capacitive impedance from dead cells.  We have changed introduction of revised manuscript as follows:

In this work we present an approach, namely impedance spectroscopy with impedance biochips with ring top electrodes, that is suitable for fast screening of bacterial cell viability (1 min). As an example, we analyse the measured change of the complex, frequency-dependent impedance Z (w) of the impedance biochip without and with up to 5 µl cell suspension with dead and live Lysinibacillus (L.) JG-A12 cells added into the region of the ring top electrode. We show that the real part of the effective impedance (ReZ (w)) and that the imaginary part of the effective impedance can be related with the thermal movement and with the membrane potential of the cells, respectively.

-->

In this work we present an approach, namely impedance spectroscopy with impedance biochips with ring top electrodes, that is suitable for fast screening of bacterial cell viability (1 min). As an example, we analyse the measured change of the complex, frequency-dependent impedance Z (w) of the impedance biochip without and with up to 5 µl cell suspension with dead and live Lysinibacillus (L.) sphaericus JG-A12 cells added into the region of the ring top electrode with 20 µl DI water. We expect a different response of live and dead cells to the osmotic challenge in 20 µl DI water. Therefore, the corresponding change of membrane potential will not fully reflect the systematic change in the concentration of live and dead cells in the liquid added to 20 µl DI water. However, using experimental procedure the effect of the treatment of L. sphaericus JG-A12 with antibiotics on the membrane potential is much stronger in comparison to the effect of osmotic shock. We clearly reveal a more resistive and less capacitive impedance from live cells and a less resistive and more capacitive impedance from dead cells. Furthermore, we discuss that the observed change of real part of the effective impedance (ReZ (w)) and that the observed change of imaginary part of the effective impedance can be related with the thermal movement and with the membrane potential of the cells, respectively.

2) Under what conditions were the cells centrifuged (too much centrifugal acceleration could also damage the cells)?

Answer: The cells were centrifuged with only 120 rpm to avoid damage due to centrifugal acceleration.  Caption of Tab. I has been revised as follows:

Concentration of L. sphaericus JG-A12 cells after adding 0.02 mg/mL the antibiotics tetracycline and 0.1 mg/ml chloramphenicol and after incubation for 3 h and 24 h, 30°C, 120 rpm in the dark.

-->

Concentration of L. sphaericus JG-A12 cells after adding 0.02 mg/mL the antibiotics tetracycline and 0.1 mg/ml chloramphenicol and after incubation for 3 h and 24 h, 30°C, 120 rpm in the dark where no damage is expected due to centrifugal acceleration.

3) On what basis is the cell count (CFU) reported? What method was used?

Answer: The correlation between optical density at 600 nm (OD600) and cell number was achieved by cell counting under a microscope using a Neubauer counting chamber.

4) There are Latin names of microorganisms in the text that are not in italics.

Answer: All Latin names of microorganisms have been set to font style italic, namely:

  1. aureus à S. aureus
  2. coli à E. coli

Staphylococcus aureus à Staphylococcus aureus

Staphylococcus epidermidis à Staphylococcus epidermidis

Escherichia cells à Escherichia cells

5) A photo of the measuring system (measuring cell with electrode) would be a valuable addition.

Answer: We have added a photo of the measuring system as Fig. 3 in the revised manuscript and renumbered Figs. 3-9 accordingly (Fig. 3-->Fig. 4, Fig. 4-->Fig. 5, Fig. 5-->Fig. 6, Fig. 6-->Fig. 7, Fig. 7-->Fig. 8, Fig. 8-->Fig. 9, Fig. 9-->Fig. 10).

This manuscript is a resubmission of an earlier submission. The following is a list of the peer review reports and author responses from that submission.